# In utero exposure to chlordecone affects histone modifications and activates LINE-1 in cord blood

Louis Legoff[1], Shereen Cynthia D'Cruz[1], Katia Bouchekhchoukha[1], Christine Monfort[1], Christian Jaulin[2], Luc Multigner[1], Fatima Smagulova[1]

Environmental factors can induce detrimental consequences into adulthood life. In this study, we examined the epigenetic effects induced by in utero chlordecone (CD) exposure on human male cord blood as well as in blood-derived Ke-37 cell line. Genome-wide analysis of histone H3K4me3 distribution revealed that genes related to chromosome segregation, chromatin organization, and cell cycle have altered occupancy in their promoters. The affected regions were enriched in ESR1, SP family, and IKZF1 binding motifs. We also observed a global reduction in H3K9me3, markedly in repeated sequences of the genome. Decrease in H3K9me3 after CD exposure correlates with decreased methylation in LINE-1 promoters and telomere length extension. These observations on human cord blood were assessed in the Ke-37 human cell line. H3K4me3 and the expression of genes related to immune response, DNA repair, and chromatin organization, which were affected in human cord blood were also altered in CD-exposed Ke-37 cells. Our data suggest that developmental exposure to CD leads to profound changes in histone modification patterns and affects the processes controlled by them in human cord blood.

## Introduction

The Developmental Origins of Health and Disease concept postulates that exposure to certain environmental factors during critical period of development and growth may have significant consequences on an individual's short- and long-term health [1]. During embryonic stage, just after fertilization, a global reprogramming event occurs to reset and establish new cell lineages in a developing organism. The second reprogramming window occurs in germ cells during somatic-to-germline transition, when the germ cells are committed to become future oocytes or gonocytes. Exposure to toxic substances during both critical reprogramming windows leads to profound changes in somatic and germ cells. In some instances, the effects can persist in adult life and can even be transmitted to subsequent generations, see for review [2].

Epigenetic reprogramming during development is associated with a cascade of events. The combination of epigenetic marks such as DNA methylation, histone posttranslational modifications (PTMs) and binding of noncoding RNAs determines the "active" or "silent" chromatin states at particular sites. The state of the chromatin can be influenced by several internal and external factors including environmental toxicants. A recent review has documented the effects of various pollutants on either site-specific or global DNA methylation in the humans [3]. The authors summarized a large number of publications wherein the effects of several pollutants such as aflatoxin B1, arsenic, bisphenol A (BPA), polycyclic aromatic hydrocarbons, tobacco smoke, and nutritional factors on genomic DNA methylation state were measured [3]. In smoking mothers, the presence of differentially methylated CpGs in the vicinity of leucine, glutamate, and lysine rich 1 (*LEKR1*) [4] and RUNX family transcription factor 3 (*RUNX3*) [5] genes in the placenta have also been documented. It was suggested that alterations in the regulation of these genes could be implicated in preterm and low birth weight of newborns. Importantly, changes in DNA methylation were also observed in the umbilical cord blood of children whose mothers smoked during pregnancy. For example, DNA methylation changes were detected in the vicinity of genes related to oxidative stress response (aryl-hydrocarbon receptor repressor [*AHRR*]), the cytochrome P450 family 1 subfamily A member 1 (*CYP1A1*), transport (myosin IG [*MYO1G*]), metabolism (acyl-CoA synthetase medium chain family member 3 [*ACSM3*]), transcription and signaling (growth factor independent 1 transcriptional repressor [*GFI1*]), contactin associated protein 1 (*CNTNAP*), SH3 and multiple ankyrin repeat domains 2 (*SHANK2*), and tripartite motif containing 36 (*TRIM36*) [6,7]. It was suggested that changes in DNA methyltransferase enzyme activity could be a major cause for differential DNA methylation [3].

Histone-PTM mechanisms are also sensitive to exposure to environmental factors and their alterations can contribute to phenotypic changes. In animal experiments, it has been shown that histone H3 lysine 4 trimethylation (H3K4me3) was affected after prenatal in utero exposure to some pesticides such as atrazine, chlordecone [8, 9, 10, 11], or the chemical plastifier bisphenol A [12], as well as by the paternal preconceptional exposure to high-fat diet [13]. In humans, the effects on histone PTMs alterations upon environmental exposure were less studied compared with DNA methylation because of scarce availability

[1]University of Rennes, EHESP, Inserm, Institut de Recherche en Santé, Environnement et Travail (Irset)–UMR_S 1085, Rennes, France  [2]Institut de Génétique et Développement de Rennes, Epigenetics and Cancer Group, UMR 6290 CNRS, Université Rennes 1, Rennes Cedex, France

Correspondence: fatima.smagulova@inserm.fr

of biological material. In a recent study, it has been shown that smoking during pregnancy leads to activation of chromatin sites that correlated with an increase in H3K4me1 and H3K27Ac and a decrease in H3K9me3 histone occupancies in the children's blood (14). Another study conducted in human blood samples demonstrated that environmental pollutants such as fine particulate matter (PM2.5) cause global changes in H3K27 acetylation, notably in genes involved in immune cell activation (15). Besides, exposure to particulate matter, urban air pollution also leads to reduction in repressive H3K9me3 and H3K9me2 marks in the postmortem human brain (16).

Changes in histone marks can interfere with chromatin structure and with the accessibility of transcription factors to DNA, which results in gene expression changes. H3K4me3, H3K4me1, H3K27Ac, and H3K79me3 at the promoters and enhancers are found to be associated with transcription activation (17). Marks related to gene silencing include H3K9me2, H3K9me3, and H3K27me3, and they are normally found within 10 kb around the promoters (17). Silencing marks such as H3K9me3 are important for the formation of pericentromeric chromatin, which is rich in repetitive DNA and plays a role in centromere functions and chromosome segregation. Besides pericentromere formation, the silencing histone marks, together with DNA methylation, are also involved in the control of transposable elements, which comprise up to 45% of the human genome (18). Among all the transposable elements, long interspersed element-1 (LINE-1 or L1) is the most active and most abundant mammalian retrotransposon element, and it encompasses 17% of the human genome. L1 elements contain a 5′ UTR promoter region and two ORFs (ORF1 and ORF2) encoding for proteins involved in reverse transcription and retrotransposition. L1 promoter hypomethylation and increased expression have been described in several cancers, including prostate (19), liver (20), and leukemia (21). L1 promoters are hypomethylated when cells are exposed to environmental pollutants (3, 22), and the activation of retrotransposons could lead to genome rearrangements (23). For example, the suppression of tumorigenicity 18 (*ST18*) gene—a gene frequently amplified in hepatocellular carcinoma—was activated by a tumor-specific L1 insertion (24).

Besides roles in controlling gene expression and chromatin structure, epigenetic mechanisms are one of the major mechanisms supporting "cell identity." All different cell types in metazoan organisms originate from a single cell, so a set of epigenetic marks serves to establish "cell identity," and these epigenetic marks are preserved during each cell division, for review see reference 25.

Thus, the importance of epigenetic mechanisms in the establishment and maintenance of fundamental cellular processes and the fact that these mechanisms can be altered by both internal or external factors prompted us to assess whether major pollutants exposure could be a source of epigenetic deregulation.

In this study, we assessed the epigenetic effects promoted by in utero chlordecone (CD) exposure. Chlordecone (also known as Kepone) is an organochlorine insecticide with well-recognized estrogenic properties (26, 27). It was extensively used from 1973 to 1993 in the French West Indies to fight against the banana root borer. This pesticide undergoes no significant biotic or abiotic degradation in the environment (28). As a consequence, permanently polluted soils and waterways have remained the primary source of foodstuff contamination and human beings continue to be exposed to this chemical. Chlordecone is a recognized neurotoxicant, reprotoxicant, and cancerogenic substance in both rodents and humans (29, 30, 31). Epidemiological studies have shown

that prenatal exposure to CD affects development of the nervous system and child growth (32, 33). In men, CD exposure at adulthood was associated with an increased risk of prostate cancer (34). Because populations are still exposed to CD, there is an immediate urge to assess the potential deleterious effects of CD, especially on fragile developmental organism.

We hypothesized that in utero CD exposure affects the epigenetic mechanisms important for the control of gene expression and silencing. To investigate this hypothesis, we used umbilical cord blood samples obtained at birth from the TIMOUN mother–child cohort in Guadeloupe and applied improved ChIP-seq techniques. We analyzed only males in our study as in utero exposed boys have developmental problems related to fine motor skills (32) and men exposed during adult life have a high risk of prostate cancer development (34). The data from human study were complemented with experimental work using the human Ke-37 cell line to identify the mechanism of action involved.

Our data show that in utero exposure to CD leads to a decrease in the amount of H3K4me3 and H3K9me3 notably in genes involved in chromosome segregation, chromatin organization, and cell cycle–related functions. CD-exposed blood cells also display a decrease in H3K9me3 in satellite DNA. The decrease in H3K9me3 was associated with decreased L1 promoter DNA methylation, altered H3K4me3 at some L1s promoters, and with telomere elongation. Our data suggest that in utero CD exposure affects epigenetic marks and thereby impacts the downstream processes controlled by epigenetic mechanisms.

## Results

A graphical overview of the experiments performed in this study is provided in Fig S1. Human cord blood samples were obtained from the mother–child Timoun cohort (see the Materials and Methods section). We analyzed histone H3K4me3 occupancy at the genome-wide level, telomere length copy number variations and LINE L1 DNA methylation changes by using genomic DNA from cord blood samples. However, for these human cord blood studies, given the limited amount of material, global amounts of H3K4me3, H3K9me3, and H4Ac were determined by the immunofluorescence method using structurally preserved nuclei (SPN) slides.

Human cord blood studies were complemented with analysis of the effects of in vitro CD exposure on the Ke-37 cell line, which originated from a human T-cell leukemia patient. In this cell line, global histone H3K4me3, H3K9me3, and H4Ac relative levels and gene expression were assessed using two doses of CD (100 and 300 nM). Nuclear ESR1 abundance was determined using immunofluorescence methods, and its binding at selected promoters was analyzed using ChIP-qPCR.

### Distribution of histone modifications are affected in newborns exposed to CD in utero

We first examined modified histone levels for modifications involved in the control of DNA repair (H4 hyperacetylation and H4Ac), transcription (H3K4me3), and epigenetic silencing (H3K9me3) in cord blood cells. We prepared SPN from umbilical cord blood and

performed quantitative analysis of immunostaining images as described in the Materials and Methods section. We found no significant changes in the levels of H4Ac, but we observed a 15% decrease in H3K4me3 and a 21% decrease for H3K9me3 levels in CD-exposed groups (Fig 1).

H3K4me3 is a pivotal mark of active chromatin; it acts as a platform for the binding of chromatin factors and nucleosome-remodeling complexes (35, 36). To reveal the possible effects of in utero CD exposure on the regulation of gene expression, we analyzed the genome-wide distribution of H3K4me3 marks using chromatin from the umbilical cord blood using six biological replicates for each group (the details are provided in the Materials and Methods section). Notably, the identified peaks had Spearman's correlation score in the range 0.95–0.99 (Fig S2), signifying that the experiments were technically reproducible. We observed that global H3K4me3 occupancy has slightly decreased in CD-exposed newborns compared with nonexposed (Fig S3), which is

consistent with our immunofluorescence data (Fig 1). Using the peak calling program MACS (with stringency $P$-value 10−5), we detected on average 24,644 peaks in control and 24,536 peaks in the CD group.

Using cutoff values fold change (FC) ≥ 1.5 and false dicovery rate (FDR) < 0.1, we determined that 706 peaks were differential between controls and exposed newborns (Table S1); details of analysis is provided in the Materials and Methods section. For example, we detected an increased H3K4me3 occupancy at ret proto-oncogene (*RET*) and reduced occupancy at BRCA1 DNA repair–associated (*BRCA1*) genes (Fig 2A). Of the differential peaks, 673 peaks were decreased in CD-exposed newborns, and 33 were found to be increased. Among the differential H3K4me3 marks, 593 were located inside coding genes (±2 kb around coding sequences); this comprised 2.5% of the total identified peaks (Figs 2B and S4A and B). We detected a significant overrepresentation of very broad peaks (>4 kb peaks) within differential peaks ($P$-value = 2.52 × 10−12;

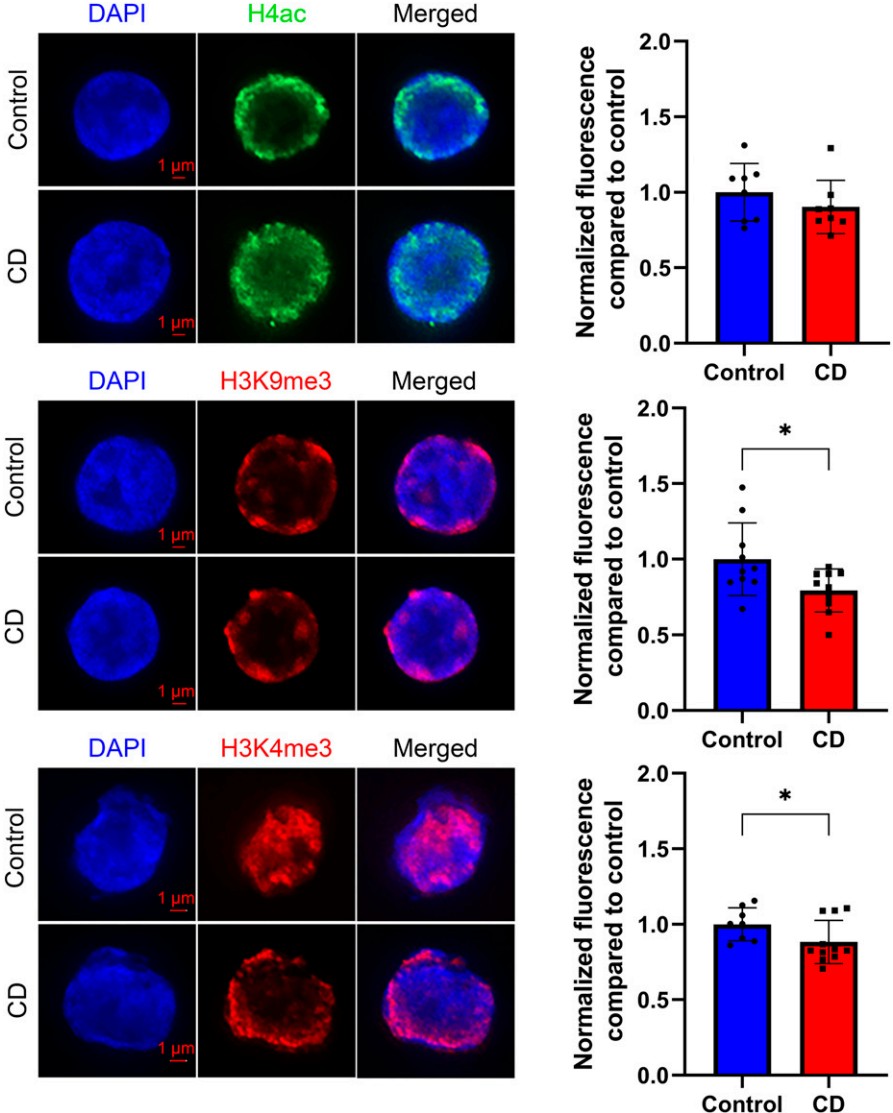

**Figure 1. H3K4me3 and H3K9me3 histone marks were decreased in CD-exposed umbilical cord blood cells.**
Representative images of structurally preserved nuclei stained with anti-H4ac (upper panel), anti-H3K9me3 (middle panel), or anti-H3K4me3 (lower panel) antibodies. Pictures were taken with fixed exposure time at ×63 objective. Corresponding quantitative analysis of the nuclear specific signal is presented as relative fluorescence compared with control ± SD. *$P$ < 0.05, Wilcoxon–Mann–Whitney test, n = 8 for each group (H4ac), n = 10 for each group (H3K9me3), n = 8 for control, and n = 10 for CD (H3K4me3).

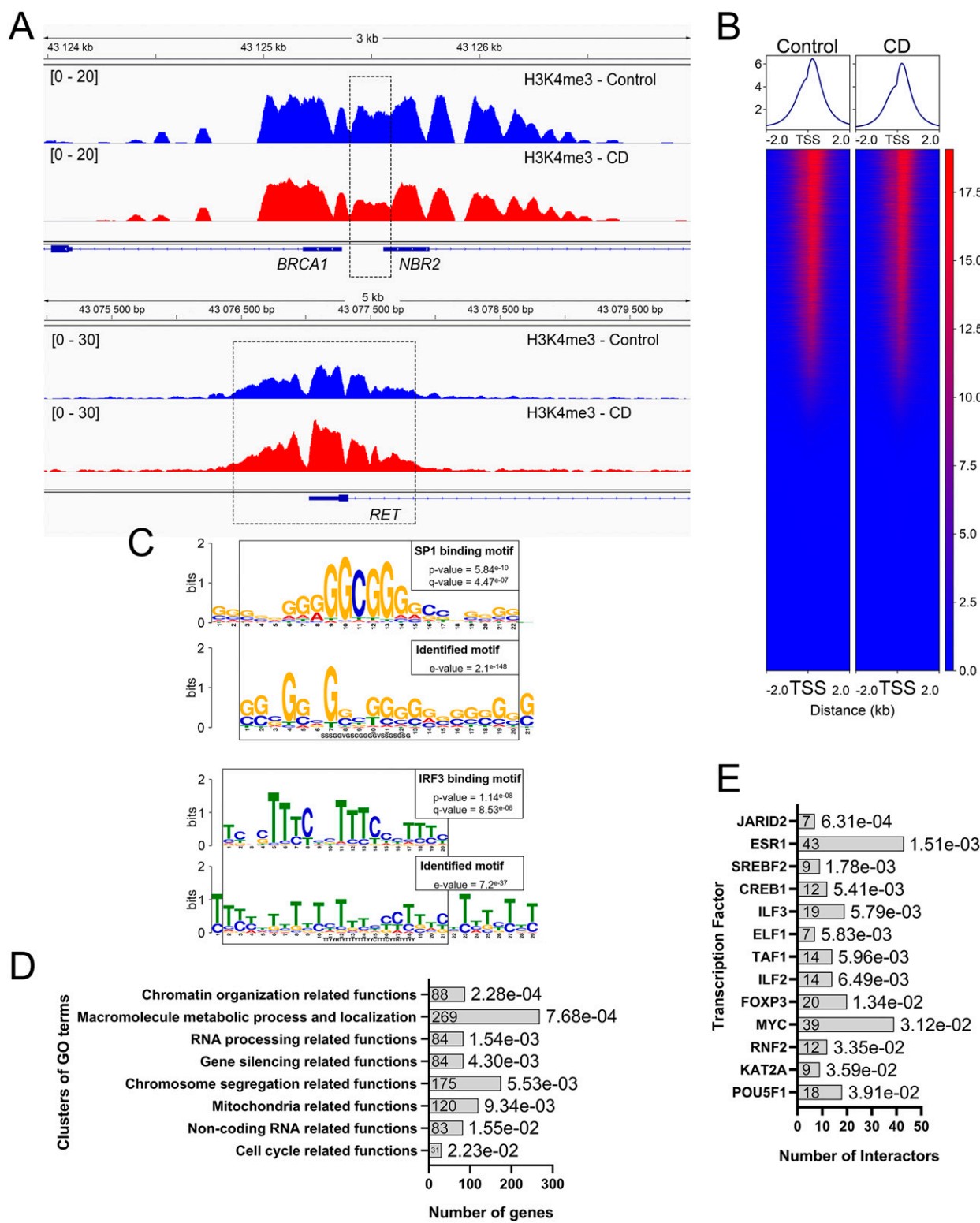

**Figure 2. The occupancy of H3K4me3 is globally decreased in CD-exposed umbilical cord blood cells.**
**(A)** H3K4me3 occupancy has decreased in the promoter of *BRCA1*. In contrast, it has increased in the vicinity of proto-oncogene *RET*. The image represents the averaged signal of six replicates. Dashed boxes indicate the position of strongest changes. **(B)** Most differential peaks are located near TSS ± 2 kb. **(C)** Differential peaks were enriched in a G-rich (upper panel) and a T-rich (low panel) motif. The parts of the motifs were similar to the sequences of SP family and IRF3 binding sites, respectively. **(D)** Several biological functions are overrepresented within the genes located in the neighborhood of the differential genes. Gene Ontology clusters are sorted by *P*-values. Adjusted *P*-values (hypergeometric test, FDR) are indicated at the end of each bar of the histogram. **(E)** Differential peaks are enriched in the genes regulated by JARID2 and ESR1 proteins.

hypergeometric test). These broad differential peaks include genes with transcription factor functions, such as AT-rich interaction domain 2 (ARID2), AT-rich interaction domain 5B (ARID5B), ETS proto-oncogene 1, transcription factor (ETS1), zinc finger protein 131 (ZNF131), nuclear receptor binding factor 2 (NRBF2), and genes with chromatin factor functions, such as chromodomain helicase DNA-binding protein 2 (CHD2) and lysine methyltransferase 2E (KMT2E). In the regions showing increased H3K4me3 occupancy, many genes encoded developmental transcription factors, for example, Wnt family member 4 (WNT4), SRY-box transcription factor 18 (SOX18), and jagged canonical Notch ligand 2 (JAG2) among others.

To address whether regulatory DNA motifs are present within the regions containing altered peaks, we used MEME-CHIP (37). We found significantly enriched motifs among the differential peaks (Fig 2C). The first discovered motif sequence is similar to the SP family of transcription factor consensus binding site, and part of the second motif is similar to interferon regulatory factor 3 (IRF3)–binding sites (Fig 2C). MEME-CHIP detected 272 target genes containing SP-binding sites including the above mentioned differential transcriptional factors (SOX18, WNT4, and JAG2), genes involved in DNA damage response (X-linked inhibitor of apoptosis [XIAP], denticleless E3 ubiquitin protein ligase homolog [DTL], and SNW domain containing 1 [SNW1]), and chromatin remodeling genes (RB binding protein 4, chromatin remodeling factor [RBBP4], SWI/SNF-related, matrix-associated actin-dependent regulator of chromatin, subfamily a, containing DEAD/H box [SMARCAD1]). Almost all genes with an increase in H3K4me3 contain binding sites for SP family members.

Next, we asked whether differential H3K4me3 regions were located near genes that share common biological functions. We assigned genes within differential peaks using GREAT (details and analysis parameters are given in the Materials and Methods section). 954 genes were enriched for several Gene Ontology (GO) clusters, biological functions, molecular processes, or cellular components (Fig 2D). Interestingly, this analysis revealed GO clusters containing genes that were related to chromosome segregation, chromatin organization, and cell cycle functions (Fig 2D and Table S2). For example, the enriched cluster "chromatin regulation related functions" contains genes such as AT-rich interaction domain 2 (ARID2), centromere protein C (CENPC), chromo-domain helicase DNA-binding protein 2 (CHD2), and stromal antigen 2 (STAG2).

Next, we compared the genes identified in differential regions using Enrichr webtool, which identify the protein–protein interactions based on previously described experimental datasets. The analysis revealed that the targets of jumonji AT-rich interactive domain 2 (JARID2) and estrogen receptor 1 (ESR1) were significantly enriched in our differential peaks (Fig 2E). For example, ESR1 has 43 targets in our differential peaks (adj. P-value = $1.51 \times 10^{-03}$), out of the 871 targets identified in the protein–protein interactions database. Among the targets of ESR1 that are found in the vicinity of differential peaks, there are genes involved in chromosome organization such as H2B clustered histone 5 (H2BC5), lysine demethylase 5A (KDM5A), X-ray repair cross complementing 5 (XRCC5), or the DNA repair gene BRCA1 (Table S3).

To experimentally validate our analysis, we studied ESR1 expression using human Ke-37 cells treated with 100 nM of CD. The cells were fixed on slides and analyzed using immunofluorescence techniques. We found that ESR1 appears as dots in the nucleus of the cells. We performed z-stacks and the quantitative analysis for each of 21 images

per stack (Fig 3A). Our analysis revealed that ESR1 nuclear staining had increased by 45% in CD-exposed Ke-37 cells compared with control (Fig 3B), suggesting that exposure to CD activates ESR1 expression. To reveal whether ESR1 could directly bind to the promoters of its target genes, we performed ChIP-qPCR analysis of some of these targets. To this end, we first identified the ESR1 binding sites using Find Individual Motif Occurrence. In total, 52,590 targets were identified in the human genome hg38 (P-value ≤ $1.0 \times 10^{-04}$). Second, we chose the gene promoters located in our differential peaks containing the ESR1-binding motif. Third, we designed ChIP-qPCR primers as described in the Materials and Methods section. For example, Find Individual Motif Occurrence analysis predicted an ESR1 binding site within the altered H3K4me3 peak located in the vicinity of lysine methyltransferase 2E (KMT2E) (Fig 3C). We performed ChIP-qPCR analysis using Ke-37 cells, and we measured 1.6 times decrease in ESR1 binding at the KMT2E promoter. Besides KMT2E, ESR1 binding has decreased in ETS proto-oncogene 1, transcription factor (ETS1), signal transducer and activator of transcription 2 (STAT2), and E3 ubiquitin protein ligase RAD18 genes by 1.60 and 1.41 and 1.42 times, respectively (Fig 3D).

Our immunofluorescence data showed that CD exposure leads to a global decrease in H3K9me3 levels in the nucleus (Fig 1). Therefore, we asked whether CD treatment of cultured cells can induce redistribution of H3K9me3. To this end, we performed chromatin immunoprecipitation using H3K9me3 antibody in Ke-37 cells that were exposed to CD. We analyzed the H3K9me3 occupancy at alpha-satellite (SATα) and satellite 2 (SAT2) repeats—which are enriched in centromere regions of the genome—as well as at some other targets by using qPCR on immunoprecipitated DNA. We detected decreased levels of H3K9me3 at both SAT2 and SATα (by 2.2 and 2.4 times, respectively) in immunoprecipitated DNA from cells exposed to CD (Fig 4A). However, no significant changes were observed for other genes, except at signal transducer and activator of transcription 2 (STAT2) suggesting that exposure to CD preferentially affect H3K9me3 occupancy at satellite DNA. Thus, both immunofluorescence and ChIP methods indicated that H3K9me3 amounts have decreased in exposed cells.

To sum up, our data show that exposure in utero to CD causes a global decrease in H3K4me3 occupancy, notably in the genes associated with chromatin organization, DNA repair, and cell cycle and in genes potentially regulated by ESR1, whereas H3K9me3 marks were mostly decreased in SAT2 and SATα.

## Telomere length is increased in cord blood from CD-exposed newborns

Because H3K9me3 are important for telomere function (38), we asked whether decrease in H3K9me3 after CD exposure could affect telomeres length. Telomeres are essential for protecting the end of chromosomes, and they consist of hexanucleotide repeats. The numbers of repeats vary among different tissues and telomere length is reduced upon cellular senescence (39). To determine whether telomere length is altered in CD-exposed newborns, we analyzed the average telomere copies ratio compared with a single-copy gene (T/S) in genomic DNA extracted from umbilical cord blood using qPCR as described in the Materials and Methods section. We found that T/S is significantly increased by 1.7 times in exposed samples compared with controls (Fig 4B). To reveal the potential candidate genes involved in telomere length changes, we

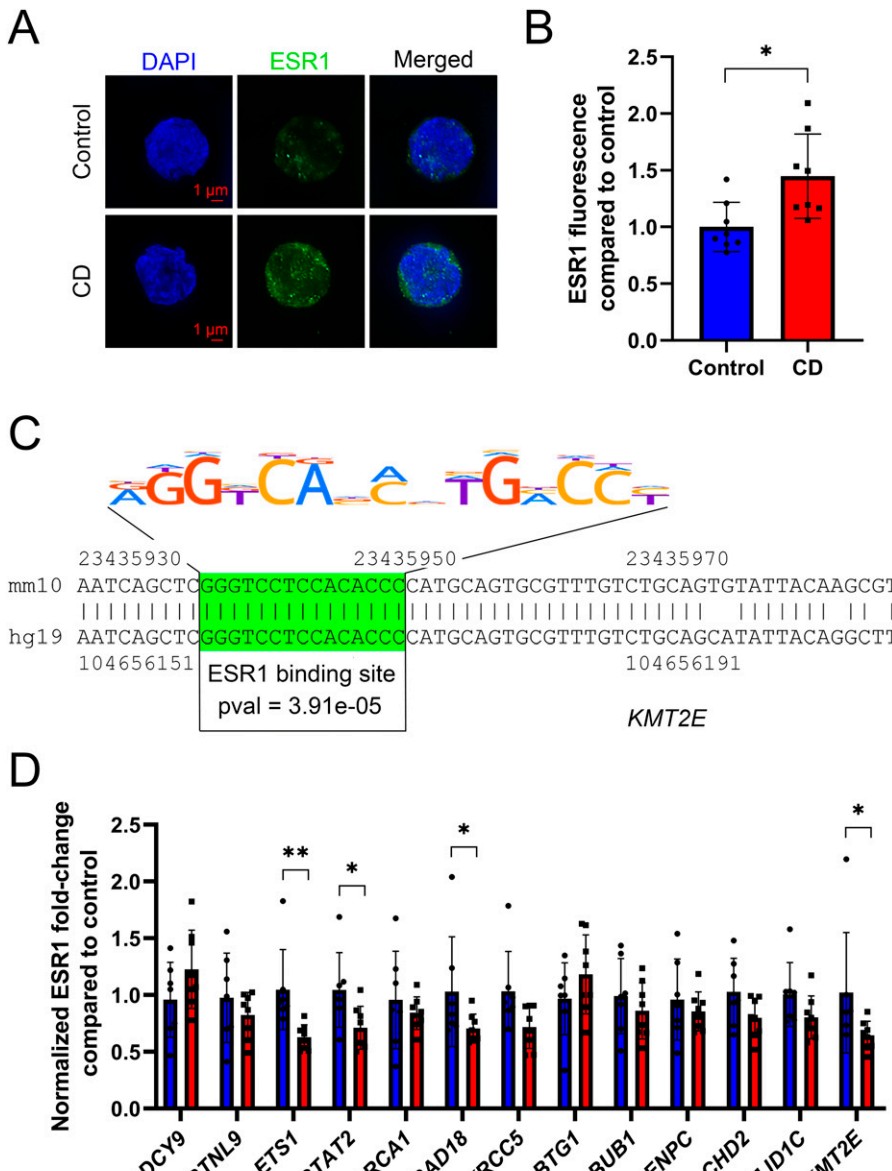

**Figure 3. ESR1 binding was altered in CD-exposed cells.**
**(A)** Representative images of z-stack sums obtained from structurally preserved nuclei immunostained with ESR1 antibody. Images were taken with fixed exposure time at x63 objective. ESR1 (green) is localized in the nucleus (visualized by DAPI staining—blue). **(B)** Quantitative analysis of ESR1 signal in nuclei was performed using ImageJ software. Data are presented as relative fluorescence compared with control ± SD. *P < 0.05, Wilcoxon–Mann–Whitney test, and n = 8 for each group. **(C)** Representation of the conservative ESR1 binding site within the *KMT2E* gene. The binding site is highlighted in green. **(D)** ESR1 binding at the promoter of target genes was assessed by ChIP-qPCR. Values were normalized to the *RPLP0* region. Control samples are in blue and CD-treated samples are in red. The data were averaged and presented as normalized ChIP fold change compared with control ± SD. *P < 0.05, **P < 0.01, Wilcoxon–Mann–Whitney test, n = 7 for control, and n = 8 for CD.

extracted all the telomere maintenance function target genes from AMIGO database. 145 genes are relevant to telomere maintenance function (GO:0000723). We found that 15 genes in our datasets have altered H3K4me3 occupancy including ATM serine/threonine kinase (*ATM*), protection of telomeres 1 (*POT1*), cAMP-dependent protein kinase inhibitor beta (*PKIB*), DNA primase subunit 1 (*PRIM1*), DNA primase subunit 2 (*PRIM2*), replication factor C subunit 3 (*RFC3*), *XRCC5*, BRCA2 DNA repair–associated (*BRCA2*) and 5'-3' exoribonuclease 1 (*XRN1*) genes (Fig 4C). Whereas the occupancy of H3K4me3 in the vicinity of these genes globally decreased in CD-exposed neonates, H3K4me3 signal had increased at the promoter of telomerase reverse transcriptase (*TERT*) (Fig S5A); however, the value did not pass the statistical significance cutoff (FC = 1.3, *P*-value

= 0.1) (Fig S5B). To determine whether telomere length was also affected in vitro, we performed the analysis of telomere length in CD-exposed Ke-37 cells. We did not observe significant differences in relative telomere length neither at low nor at high dose of CD (Fig S6), suggesting that short-term exposure to CD in our experimental conditions does not induce telomere length change. However, our data suggest that telomere length was elongated in cord blood cells because of long-term CD exposure.

**LINE-1 activation in CD-exposed newborns**

Because we observed a decrease in H3K9me3 amounts, we asked whether retroelements activity could be induced in CD-exposed cells. To

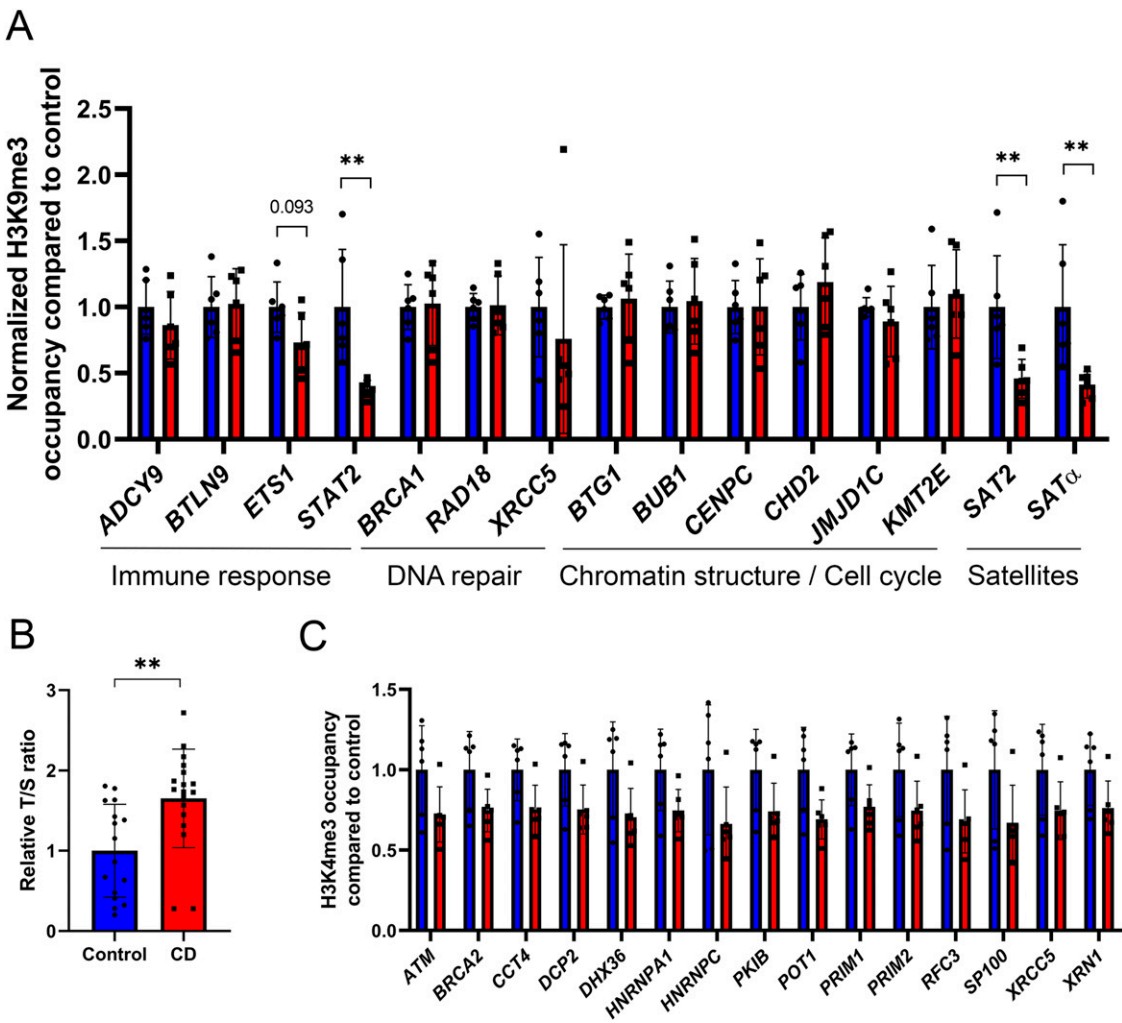

**Figure 4. H3K9me3 occupancy and telomere length changes in CD-exposed umbilical cord blood.**
**(A)** H3K9me3 occupancy at the promoter of target genes and satellite was assessed by ChIP-qPCR. Values were normalized to the *RPLP0* region. Control samples are in blue, and CD-treated samples are in red. The data were averaged and presented as normalized ChIP fold change compared with control ± SD. *$P < 0.05$, **$P < 0.01$, Wilcoxon–Mann–Whitney test, and n = 6 for each group. **(B)** Telomere length was assessed by qRT-PCR and normalized to the qPCR value of a single-copy gene (*RPLP0*) to determine the T/S ratio. **$P < 0.01$, Wilcoxon–Mann–Whitney test, n = 16 for control, and n = 18 for CD. **(C)** Histone occupancy of H3K4me3 at the promoter of genes involved in telomere maintenance function. Control samples are in blue and CD-treated samples are in red. Peak intensities were extracted from ChIP-seq data and were plotted as fold change compared with control ± SD. All plotted regions displayed statistical difference in H3K4me3 occupancy (moderated *t* test, Limma package), n = 6 for each group.

determine whether methylation of L1 promoters could be altered by CD exposure, we performed qPCR on *Hpa*II enzymatically digested and undigested genomic DNA from umbilical cord blood. For the analysis of L1 promoter methylation, we chose previously published sequence of L1.12 element (accession number U93565), the sequence of the promoter we studied (354 bp) is presented in Fig 5A. Blast analysis of this promoter compared with the human reference genome showed that this sequence is present 116 times (alignment length ≥ 350 bp, % of identity >95%, maximum five mismatches). Out of the three CpGs tested, we observed that methylation of $CpG_3$ was significantly decreased by 21.3% in CD-exposed samples compared with control, *$P < 0.05$ (Fig 5A), suggesting that CD exposure could lead to decreased methylation at least at some targets.

Next, we asked whether DNA methylation at CpG is associated with changes in H3K4me3 in L1s of CD-exposed newborns. To this end, we

extracted coordinates of full-length L1s using L1Base (v2), a database containing putatively active LINE-1 (http://l1base.charite.de/index.php). We remapped the sequencing reads of H3K4me3 to the reference human genome without filtering multimapped reads (i.e., reads mapping to multiple positions on the genome). We detected 186 L1s having H3K4me3 peaks out of 13,671 L1s. Of these 186 L1 elements, 90 have altered H3K4me3 occupancy in the CD group compared with control (FC ≥ 1.2, 22 increased, 78 decreased, Table S4). For example, we detected increased H3K4me3 at the promoter of UID-34 L1, retroelement is located on chromosome 19 between the *LINC01801* and *LINC00904* genes (Fig 5B). The promoter of UID-34 contains three CpGs, in which the DNA methylation was analyzed and described above. The decrease in DNA methylation in $CpG_3$ is associated with an increase in H3K4me3, suggesting possible activation of transcription of UID-34 retroelement. Interestingly, within L1 promoters with decreased H3K4me3, we detected

Figure 5. **Exposure to low dose of CD leads to DNA methylation reduction and increased H3K4me3 at the UID-34 L1 in umbilical cord blood cells and increased expression of LINE-1 in Ke-37 cells.**
**(A)** Methylation analysis was performed by qPCR analysis using the intact cord blood genomic DNA or digested by methylation-sensitive restriction enzyme from control samples (in blue) and CD-treated samples (in red). Data are presented as relative methylation compared with control ± SD. *P < 0.05, Wilcoxon–Mann–Whitney test, and n = 8 for each group. Below the graph, the sequence of the L1.12 promoter with CpG positions is shown. **(B)** H3K4me3 occupancy at the promoter of UID-34 L1. The image represents the averaged signal of six replicates. Dashed boxes indicate the position of the strongest changes. CpGs positions are indicated. **(C)** H3K4me3 occupancy at CpG3 of the L1 promoter. Values were normalized to the *RPLP0* region. The data were averaged and presented as normalized ChIP fold change compared with control ± SD. *P* = 0.1, Wilcoxon–Mann–Whitney test, and n = 7 for each group. **(D)** The expressions of LINE-1 ORF1 and ORF2 in Ke-37 cells were assessed by qRT-PCR following exposure to 100 nM of CD. **(E)** The expressions of LINE-1 ORF1 and ORF2 at 300 nM of CD. The expression of target genes was normalized to the expression of the housekeeping *RPL37A* gene. The data were averaged and presented as normalized expression fold change compared with control ± SD. **P < 0.01, ***P < 0.001, Wilcoxon–Mann–Whitney test, n = 18 for control, n = 20 for CD (100 nM), and n = 6 for each group (300 nM).

an overlap with protein-coding gene promoters (such as at the dual promoter of *RABL3*/*GTF2E1*) see Table S4.

Next, we analyzed H3K4me3 occupancy in the vicinity of CpG3 using ChIP on Ke-37 cells exposed or not to 100 nM of CD. We detected a 16% H3K4me3 increase in treated cells compared with control; however, the difference did not pass the significance cutoff (exact *P*-value = 0.08) (Fig 5C). To determine whether the expression of L1 was altered by CD exposure, we performed qRT-PCR on RNAs

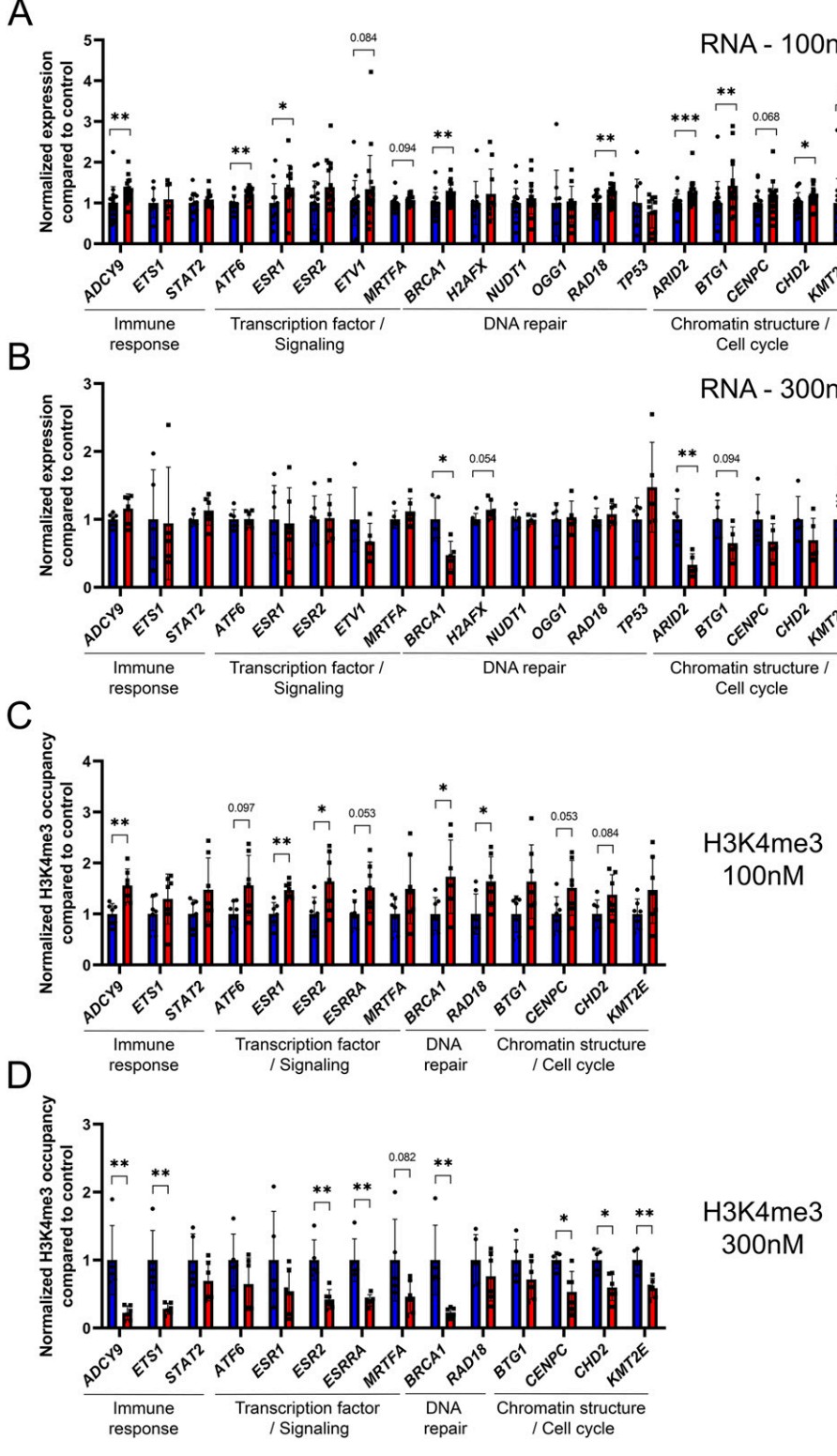

**Figure 6. Gene expression changes were associated with H3K4me3 occupancy alterations in CD-exposed Ke-37 cells.**
**(A, B)** Gene expression in Ke-37 cells was assessed by qRT-PCR after exposure to 100 nM (A) or 300 nM (B) of CD. The expression of target genes was normalized to the expression of the housekeeping *RPL37A* gene. The data were averaged and presented as normalized expression fold change compared with control ± SD. *$P < 0.05$, **$P < 0.01$, ***$P < 0.001$, Wilcoxon–Mann–Whitney test, n = 18 for control, n = 20 for CD (100 nM), and n = 6 each group (300 nM). **(C, D)** H3K4me3 occupancy at the promoter of target genes in Ke-37 cells was assessed by ChIP-qPCR after exposure to 100 (C) or 300 nM (D) of CD. Values were normalized to the *RPLP0* region. The data were averaged and presented as normalized ChIP fold change compared with control ± SD. *$P < 0.05$, **$P < 0.01$, Wilcoxon–Mann–Whitney test, n = 7 for each group (100 nM), n = 5 for control, and n = 6 for CD (300 nM).

extracted from Ke-37 cells. The analysis revealed that both ORFs of L1 were significantly overexpressed by 1.5 and 1.3 times, respectively, following exposure to 100 nM of CD (Fig 5D). The exposure to 300 nM of CD did not show any significant change in

L1 expression (Fig 5E). Our data suggest that exposure to CD leads to increased transcription of LINE-1 retrotransposon sequences that could result from decreased methylation at least in some promoters.

### Gene expression and histone H3K4me3 occupancies increased in CD-exposed Ke-37 cells at a 100 nM dose and decreased at a 300 nM dose

To confirm the results that we have observed on umbilical cord blood, we performed in vitro analysis using the cultured cell line Ke-37. First, we asked whether exposure to CD affects gene expression at targets where H3K4me3 was affected in umbilical cord blood. We exposed Ke-37 cells to two different doses of CD, 100 and 300 nM. We measured the expression of genes encoding DNA repair, chromatin remodeling, and transcription factors. Surprisingly, we found that exposure to low and high doses led to different results. Low-dose exposures led to gene expression increase, whereas high-dose exposure led to a decrease in gene expression. For example, at low dose, we observed 1.2 times increase in transcription of the DNA repair gene *BRCA1*, 1.3 times in the adenylate cyclase 9 (*ADCY9*) gene, 1.2 times in chromatin remodeling genes *KMT2E* and *ARID2*, respectively, and 1.4 times in transcriptional factor *ESR1* (Fig 6A). In contrast, at high dose, the observed changes were opposite, with expression being decreased in *BRCA1* 2.1 times; *ARID2*, 3.0 times; *KMT2E*, 2.0 times (Fig 6B).

To address whether these changes are associated with histone H3K4me3 occupancy changes, we performed ChIP-qPCR and found that in the majority of the cases, the changes in RNA levels are correlated with alterations in histone H3K4me3 occupancy (Fig 6C and D). For example, the expression of *BRCA1* and the histone occupancy at its promoter has consistently increased at dose "100 nM" 1.2 and 1.7 times, respectively, and decreased at dose "300 nM" 2.1 and 4.3 times, respectively. To address whether histone marks H3K4me3 and H3K9me3 were affected in Ke-37, we analyzed these histone marks by immunostaining SPN. Quantitative analysis revealed that exposure to 300 nM of CD leads to 1.9 times decrease in H3K4me3 levels compared with control. No significant difference was observed for H3K9me3 levels (Fig S7).

Thus, our analysis shows that exposure of cultured cells to CD leads to gene activation at low dose and repression at high dose for targets which were previously identified in human samples.

## Discussion

In animal models, it has been shown that exposure to CD affects germ cells in both sexes (9, 11) and causes global alterations in somatic tissues such as prostate and liver (40). This study is designed to evaluate the effects of CD on humans by using cord blood obtained from males. How is it relevant to use umbilical cord blood for analysis of prenatal effects of chlordecone on its target tissue? In a previous study, a strong correlation between gestational bisphenol A (BPA) exposure and brain-derived neurotrophic factor (*BDNF*) DNA methylation in the umbilical cord blood was observed in the children from Columbia Center for Children's Environmental Health cohort (41). The DNA methylation profile at promoter was correlated in blood and brain as was proved by analysis of the postmortem brain (42). The CD exposure occurred during entire early embryonic period, including early embryonic stage, when the tissue layers start to form, so common progenitors of many tissues could be affected. One could imagine that the promoted effects could be observed in many tissues, including blood, similar to BPA exposure.

We found that exposure to CD leads to global decrease in H3K4me3 occupancy in human cord blood and in Ke-37 cells exposed to high doses of CD. We previously observed such a decrease in gene expression in murine embryonic testis exposed to low dose of CD (9), suggesting that CD exposure affects transcription. Similar to CD, exposure to another estrogenic compound, diethylstilbestrol, also led to a decrease in DNA repair genes' expression (43). We reported that female mice exposed to CD display an excess of 8-oxoguanine in exposed embryonic ovaries and γH2AX signals were detected in adult ovaries, suggesting a genotoxic property for CD, perhaps via the generation of an oxidative stress (11). We believe that the global decrease in histone marks in this study might be the consequence of a cellular response to CD-induced genotoxic stress. Indeed, it has been shown that under toxic conditions the transcription is generally globally reduced (44).

Among the most profound changes in cord blood, genes involved in chromosome segregation, chromatin organization, cell cycle, and DNA repair were detected. For example, we found a decrease in H3K4me3 occupancy in the *BRCA1* gene in human cord blood and in Ke-37 cells exposed to a high CD dose. Our observation is consistent with previous studies where it was shown that DNA damaging agents induce significant decrease in *BRCA1* and *BRCA2* mRNA levels (45). Because cells defective in *BRCA1* show genomic instability, telomere dysfunction (46), and increased telomerase activity (47), we believe that decrease in *BRCA1* in CD-exposed samples could eventually lead to genomic instability.

Notably, we found a decreased intensity of broad H3K4me3 peaks. In general, these peaks are located in key genome maintenance genes. Decrease in peak intensity is associated with repression of tumor suppressor activity (48). For example, we observed a H3K4me3 peak decrease at the tumor suppressor gene *KMT2E* that is frequently deleted in myeloid malignancies; the product of this gene controls cell cycle and genomic stability and regulates hematopoiesis (49). Thus, we suggest that decrease in H3K4me3 at the *KMT2E* gene promoter after CD exposure could also contribute to a higher risk of genetic instability.

We observed a large number of ESR1 potential targets in our differential peaks. We previously determined that exposure to CD leads to an increase in both protein and RNA levels of ESR1 and affects the binding of ESR1 to its targets in embryonic testis (9). To date, the role of ESR1 in blood cells is not well documented. There is evidence that lymphocytes and monocytes are responsive to estrogens (50). Moreover, it was reported that exposure to estrogen is likely to limit the efficiency of viral emergence from latency in blood cells, suggesting that ESR1 could modulate immune response (51). Considering the CD-induced alteration of several ESR1 target genes involved in immune response (interleukin enhancer-binding factor 2 [*ILF2*], CD53 molecule [*CD53*], and calcium/calmodulin-dependent protein kinase IV [*CAMK4*]) in our ChIP-seq data (Table S1), it is conceivable that developmental exposure to CD could alter the sensitivity to infections. However, the detailed effects of CD on immune response remain to be investigated.

Besides immune response, ESR1 could be involved in the alteration of telomere length that we have observed. We noted that among ESR potential targets, several genes are essential for telomere maintenance function: *XRCC5*, chaperonin-containing TCP1 subunit

4 (*CCT4*), and tumor protein p53 (*TP53*). Moreover, in men, a high estradiol level is associated with longer telomere length (52). Because an estrogen response element is present in the promoter of the telomerase gene (*TERT*), it was proposed that estrogens could directly stimulate telomerase expression (53). Estrogen-induced increased telomerase activity was demonstrated in several cell lines (54, 55, 56). Although telomerase activity is repressed in most somatic tissues, in highly proliferative tissues such as hematopoietic system, digestive system, and skin, *TERT* is still expressed (57), which makes possible the regulation of the *TERT* gene by estrogens (58).

On the other hand, histone trimethylation on H3K9 and H4K20 in telomeric regions could also regulate telomere lengthening (59). Since telomere elongation was found to be associated with decreased histone H3K9 trimethylation (60, 61), it is also conceivable that the observed increase in telomere length in exposed samples is a consequence of CD-induced H3K9me3 decrease. We observed that H3K9me3 occupancy has decreased in satellite regions. We believe that it could affect the transcription at centromeric satellite DNAs. Overexpression of centromeric transcripts leads to chromosome missegregation, reviewed in reference 62. Notably, we observed that decrease in H3K9me3 is also associated with DNA methylation decrease in L1 retrotransposon elements. Our data are consistent with several studies showing that exposure to environmental toxicants leads to LINE-1 methylation decrease. Remarkably, changes in the methylation state of retroelements can be caused by exposure to different compounds, such as arsenic (63), cadmium (64), as well as air pollution (65). It is suggested that oxidative DNA damage affects the capacity of methyltransferases to interact with DNA, resulting in lower methylation levels at CpG sites (66). Many studies have shown that L1s are involved in genetic rearrangements and somatic L1 insertions were detected in some types of human cancers, such as colon cancer (67). To explain the strong correlation between *TP53* mutation and LINE-1 expression, it was suggested that its product, p53, could play a role in L1 repression (67). Because we observed reduced H3K4me3 at the *TP53* gene, it is conceivable that impaired regulation of *TP53* could contribute to L1 expression changes. L1s become more active in somatic tissues during the course of aging (68). Thus, decreased methylation at L1 could also contribute to premature cellular senescence in spite of telomeres elongation. The decreased LINE-1 DNA methylation correlates with an increased H3K4me3 occupancy at L1 promoters. The changes in DNA methylation could lead to altered chromatin conformation as DNA methylation and histone modifications are functionally linked.

We found that exposure to low and high CD doses leads to different responses in terms of H3K4me3 occupancy and gene expression. The most plausible explanation is that a low-dose exposure possibly reflects the most specific answer to CD. In contrast, high-dose exposure would lead to a more general toxic effect. Some differences between cell line and blood cell CD-induced effects could be explained by the fact that the cell composition in umbilical cord is not the same as in cell line. Besides that, in vivo samples have influence from other organs, for example, hormones and secreted factors from hormonal tissues.

## Conclusion

Our data suggest that exposure to chlordecone leads to alterations in epigenetic marks that regulate gene activation and silencing.

Dysregulation in those mechanisms may increase the risk of genetic instability.

# Materials and Methods

### Human cord samples

The TIMOUN study is a prospective mother–child cohort study from the Guadeloupe archipelago in the French West Indies. Between November 2004 and December 2007, 1,068 pregnant women were enrolled during third-trimester check-up visits at public and private health centers (69). At delivery, umbilical cord blood samples were obtained and collected into spray-coated EDTA tubes. Centrifuged plasma samples and whole blood samples were stored for further analysis. Plasma chlordecone concentration was determined by high-resolution gas chromatography and Ni63 electron capture detection. Detailed information about sampling, analysis, and quality assurance and control has been provided elsewhere (34, 70). After analysis of the quality control samples consisting of human plasma spiked with a series of concentrations of chlordecone, we defined the limit of detection (LOD) for plasma chlordecone as 0.02 ng/ml. From the 344 male umbilical cord blood samples available (chlordecone range < LOD to 12.47 ng/ml; median: 0.20 ng/ml; Fig S8), we selected 16 with the highest chlordecone values (range: 0.83–12.47 ng/ml; median: 1.15 ng/ml) and 16 were randomly selected among samples with less than LOD values. They were referred to as the exposed and control groups, respectively.

Whole umbilical cord blood samples were used for genomic DNA extraction from 1 ml of blood. Briefly, DNA was treated with proteinase K and RNAse A and was loaded on a spin column (DNeasy Blood and Tissue Kit; 69506; QIAGEN). After subsequent several washes, DNA was eluted and the quantity of DNA was estimated using nanodrop to verify the A260/A280 and A230/A260 ratios. In addition, the concentration was verified by a fluorescence method using the QuantiFluor ds DNA System dye (E2670; Promega). To verify DNA integrity, we ran DNA on agarose gel and we observed normal undegraded intact DNA.

SPN from umbilical cord blood for three-dimensional analysis were prepared from frozen blood by several times of washing in DMEM medium (Life Technologies, GIBCO) with 0.5% EASYpack protease inhibitor cocktail tablets (04 693 132 001; Roche). 100 $\mu$l of blood was pelleted at 1,000 g and rinsed three times in 1 ml DMEM. The final pellet was resuspended in 200 $\mu$l DMEM. 25 $\mu$l of cell suspension was mixed with equal amounts of 3.7% (vol/vol) paraformaldehyde and 0.1 M sucrose and spread on glass slides, and the slides were air-dried and kept at –80°C.

### Cell culture

The Ke-37 cell line was cultured in RPMI 1640 supplemented with 10% FBS and antibiotics and glutamine. Two million cells were treated with 100 or 300 nM of chlordecone (sc-394278; Santa Cruz), diluted in DMSO. The dose 100 nM that was used in our study is four times higher than the highest concentration of CD found in our exposed neonate group (12 ng/ml or 24 nM). Considering that

concentration of CD in humans could reach up to 20 ng/ml or 40 nM, this dose is relevant to environmental exposure. We also opted to study the effects of CD at a higher dose (300 nM) to better understand the mechanism of CD action on human cells. Cells were harvested 18 h after exposure, spin washed in PBS, snap frozen in liquid nitrogen, and kept at −80°C until use. Slides with SPN from Ke-37 cells were prepared similarly to cord blood samples, except only one wash in PBS was performed prior to fixation.

### Immunofluorescence

Slides were washed several times with PBS before use and 2 min with 0.1 M glycine in PBS to remove traces of paraformaldehyde. Slides were permeabilized during 30 min in PBS/0.5%Triton at RT, washed with PBS, and incubated for 30 min in blocking solution (0.1% [vol/vol] donkey serum, 0.03% [wt/vol] BSA, and 0.005% [vol/vol] Triton X-100 in PBS). SPN were immunostained with rabbit polyclonal H4Ac (06-946 1:500; Millipore), or rabbit polyclonal H3K4me3 (07-473, 1:500; Millipore) or rabbit polyclonal H3K9me3 (ab8898, 1:500; Abcam) or rabbit polyclonal ESR1 (ab32063, 1:50; Abcam) antibodies at 4°C overnight followed by several washes and incubation with fluorescent Alexa secondary antibodies. Z-stacks were acquired with 500 nm steps; 21 individual planes were taken for each individual channel, for DAPI (Blue, 350 nm), or histone marks (Green, 488 nm) using Zen Pro (version 2.3) program. All images for control and exposed samples were taken with fixed exposure time. For immunofluorescence analysis, we chose cells with similar nucleus morphology. Considering that highest cell population are granulocyte and lymphocyte cells, our results reflect changes in these major cell populations. Deconvolution was performed using the "fast iterative" algorithm provided by Zen Pro. The mean intensity images were generated for each z-stack and the resulting images were analyzed in ImageJ v1.52n. We used the lasso tool for nucleus and cytoplasm contouring and the integrated density immunofluorescence for each nucleus was calculated. The similar area with background was used and the background was subtracted. We analyzed 12 independent biological replicates for control and treated groups and at least 15 cells for each replicate. The data are presented as corrected total cell fluorescence of normalized fluorescence compared with control ±SD. The Wilcoxon–Mann–Whitney test was used to assess the statistical significance.

### RNA extraction and qRT-PCR

Total RNA was extracted using the RNeasy Plus Mini Kit (QIAGEN), which includes the DNA elimination step. The snap-frozen Ke-37 cells were lysed, DNA eliminated, and DNA-free RNA was placed on column, washed, and eluted according to the manufacturer's instructions. For qRT-PCR analysis, a minimum of six biological replicates was used for both control and CD-exposed cells. Reverse transcription was performed with 1 $\mu$g of RNA using the iScript Reverse Transcription kit (Invitrogen) according to the manufacturer's instructions. The resulting cDNA was diluted 10 times to a final volume of 200 $\mu$l, and 4 $\mu$l was used for quantitative PCR. QPCR was performed using a Bio-Rad 384 plate machine using iTaq Universal SYBR Green Supermix (1725124; Bio-Rad). Ct values

for *RPL37A*, a housekeeping gene, were used for normalization using CFX manager software provided with Bio-Rad 384 plate machine. The primer sequences used for qPCR are shown in Table S4. The data were analyzed and are presented as mean values of FC compared with control ±SD. Minimum six replicates was used for control and exposed groups. The Wilcoxon–Mann–Whitney test was used to assess the statistical significance.

### Chromatin immunoprecipitation (ChIP), ChIP-seq, and ChIP-qPCR

ChIP from cells was performed as previously described (71) with small modifications. We performed ChIP-seq experiments using six replicates for control or CD-exposed samples. 200 $\mu$l of cord blood cells was resuspended in 10 ml of PBS, fixed in 1% paraformaldehyde for 10 min at room temperature, quenched with 0.125 M glycine, and pelleted at 900$g$ for 10 min. Centrifugation at 900$g$ during 10 min precipitate mainly intact nuclei; we verified the cell suspension under microscope, and we observed that most nuclei were intact. Partially broken nuclei were eliminated in the next step by washing with low detergent solution and sedimentation at 900$g$, the broken nuclei will not precipitate at this speed. The pellet was lysed in 300 $\mu$l lysis buffer (1% SDS, 10 mM EDTA, and 50 mM Tris–HCl pH 8), and sonication was performed using Qsonica 700 sonicator supplied with cup horn 431C2. Sonication was performed in 300 $\mu$l in Eppendorf tubes using the following conditions: efficiency 60%, sonication 20 s on, 20 s off, and total time 4 min. Average chromatin fragments size was 300 bp. The sonicated chromatin was diluted six times in ChIP dilution buffer (1.1% Triton X-100, 1.2 mM EDTA, 16.7 mM Tris–HCl, pH 8, and 167 mM NaCl) supplemented with protease inhibitors (Roche). We performed ChIP using 0.5 $\mu$g of H3K4me3 (07-473; Merck Millipore) or 0.5 $\mu$g of H3K9me3 (ab8898; Abcam), or 0.5 $\mu$g of ESR1 (ab32063; Abcam) antibodies and 20 $\mu$l of magnetic beads (Dynabeads; 10002D; Thermo Fisher Scientific). After several washes, immunoprecipitated DNA was eluted, treated with proteinase K, and purified using the QIAGEN MinElute kit (28004; QIAGEN). Sequencing libraries were prepared using NEBNext Ultra DNA Library Prep Kit for Illumina (E7645S; NEB). We used 3–5 ng DNA for library preparation, and 15 cycles were used for library amplification. Sequencing was performed on an Illumina HiSeq4000 sequencer using a single-end 50-base read in multiplexed mode. Adapter dimer reads were removed using DimerRemover. The reads were mapped to the reference genome *hg38* using Bowtie2 v2.2.7. The numbers of mapped reads were normalized by a scale factor to adjust the total number of reads. From the aligned reads, H3K4me3 peaks were identified using 12 biological replicates and the corresponding input by MACS2 (v2.1.1) algorithm; the following parameters were applied: a shift-size window of 73 bp, no model, and a $P$-value threshold $<10 \times 10^{-05}$. To compare the H3K4me3 ChIP datasets of the CD-exposed and control samples, differential peaks were identified using the following steps: first, from all of the peaks called using MACS2, we retained only peaks with average values above the median; second, we selected peaks with fold changes above 1.5 and FDR < 0.1. Statistical significance was calculated using R package Limma v3.38.3 that was designed to work with a limited number of replicates. We performed functional annotation of the differential peaks using the web-based tool GREAT v3.0.0 (default parameters). The differential peaks were visualized using the

Integrative Genomics Viewer (v2.4.2). In umbilical cord blood, there are several cell types; the granulocytes and lymphocytes are the major cell types. Because each cell type have cell-specific marker expression, it is feasible to verify the proportion of given cell type (72). For example, there are a large number of commonly used macrophage markers such as CD14, CD16, CD64, CD68, CD71, and CCR5. We detected in our ChIP-seq data that cell-specific markers for different cell types have no altered promoter H3K4me3 occupancy in control and exposed samples, suggesting that cell population is not altered by CD exposure, Fig S9.

For retroelement LINE L1 studies, genome-wide alignment and peak calling of the sequencing reads were mapped to the reference genome *hg38* using Bowtie2 v2.2.7 as described above. For the second mapping, multimapped reads were not discarded from the alignment file. For each multimapped read identified, only the best alignment was conserved. In the case of perfect multiple matches (based on the MAPQ field), one of the best alignments was randomly conserved. This corresponds to the default algorithm of bowtie2 when analyzing the multimapped reads. The numbers of mapped reads were normalized by a scale factor to adjust the total number of reads. From the aligned reads, H3K4me3 peaks were identified using 12 biological replicates and the corresponding input by MACS2 (v2.1.1) algorithm; the following parameters were applied: a shift-size window of 73 bp, no model, and a *P*-value threshold <0.05. The identified peaks were intersected with coordinates of putatively active L1s, downloaded from L1Base (v2).

For ChIP-qPCR, six biological replicates were used for Ke-37 cells treated with 300 nM of CD. For ChIP-qPCR of cells treated with 100 nM of CD, we used 18 biological replicates for control and 20 biological replicates for CD. The primer sequences used for ChIP-qPCR are shown in Table S5. QPCR was carried out as described in the previous section. The internal region (far from promoter) of the *GAPDH* or *RPLP0* gene was used for normalization. We assessed the efficiency of ChIP by analyzing ChIP/input ratio. In most of the cases, we observed an enrichment of immunoprecipitated regions compared to input. The data were analyzed and are presented as mean values of FC compared with control ±SD. The Wilcoxon–Mann–Whitney test was used to assess the statistical significance.

### MEME-CHIP

For the analysis, we used the sequences from differential peaks. Motif enrichment analysis was performed with MEME-CHIP using the default parameters. Identified motifs were compared with known motifs using TomTom (73) and the HOCOMO database v11 with a *P*-value threshold of 0.05.

### DNA methylation analysis

Methylated DNA was extracted and enriched using the EpiMark kit (E3317S; NEB) according to the protocol provided by the manufacturer. We used 500 ng for the digestion by the methylation-sensitive enzyme *Hpa*II that recognizes only unmethylated GGCC/CCGG sites. After enzyme digestion and proteinase K treatment, DNA was diluted to concentration 0.3 ng/$\mu$l. For qPCR analysis, we used 1.2 ng of DNA and primers were designed for LINE L1. Each pair of primers amplifies a region with a unique CpG site. We normalized

LINE L1 PCR with amplification of a DNA fragment that has no *Hpa*II site within the unique gene *POLR2K*. The percent of methylation was calculated as the ratio of "cut" to "uncut" product and presented as changes compared with control.

### Primer design for ChIP-qPCR (for unique regions)

To design primers for ChIP-qPCR analysis of regions found with differential H3K4me3 occupancies, we identified the coordinates of altered regions using the Integrative Genomics Viewer genome browser and bed file results from MACS2 peak calling. We extracted the coordinates of the peaks from the bed file and we used the UCSC genome browser to upload DNA genomic sequences. We masked the repeated sequences to obtain the sequences without repetitive sequences to avoid multiple priming. We designed primers near the center of the peaks (±200 bp) by using Primer-BLAST tool from the National Center for Biotechnology Information (NCBI).

## Data Availability

All sequencing and ChIP-seq data from this study are publicly available and have been deposited in the National Center for Biotechnology Information Gene Expression Omnibus. The GEO number is GSE171285 https://www.ncbi.nlm.nih.gov/geo/query/acc.cgi?acc=GSE171285.

### Ethics statement

The research procedures were approved by the Guadeloupean Ethics Committee for biomedical studies involving human subjects (Project no. 03-04 01/10/2004). Written informed consent was obtained from parents of each participating newborn.

## Supplementary Information

## Acknowledgements

Sequencing was performed by the GenomEast platform, a member of the "France Génomique" consortium (ANR-10-INBS-0009). Funding: This work was supported by funding from Atip-Avenir to F Smagulova (R13139NS). L Legoff was supported by Allocations de Recherche Doctorale (ARED)/l'Institut National de la Santé et de la Recherche Médicale (INSERM) fellowship. This work was also supported by the French National Health Directorate Grant (Grant R20024NN) to Luc Multigner. However, the funder has no role in study design, data collection and analysis, decision to publish, or preparation of the manuscript.

### Author Contributions

L Legoff: software, formal analysis, investigation, visualization, methodology, and writing—original draft.

SC D'Cruz: investigation, methodology, and writing—original draft, review, and editing.

K Bouchekhchoukha: investigation and methodology.

C Monfort: methodology and writing—original draft.

C Jaulin: methodology and writing—original draft, review, and editing.

L Multigner: resources, methodology, and writing—original draft.

F Smagulova: conceptualization, supervision, funding acquisition, investigation, methodology, project administration, and writing—original draft, review, and editing.

## Conflict of Interest Statement

The authors declare that they have no conflict of interest.

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
