## [Reviewer comments · Life Science Alliance]

Life Science Alliance

In utero exposure to chlordecone affects histone modifications and activates LINE-1 in cord blood.

Louis Legoff, Shereen D'Cruz, Katia Boucheekhchoukha, Christine Monfort, Christian Jaulin, Luc Multigner, and Fatima Smagulova

DOI: <https://doi.org/10.26508/lsa.202000944>

Corresponding author(s): Fatima Smagulova, Inserm/IrsetU1085

Review Timeline:

Submission Date:	2020-10-21
Editorial Decision:	2021-03-24
Revision Received:	2021-03-26
Accepted:	2021-03-29

Scientific Editor: Shachi Bhatt

Transaction Report:

March 24, 2021

RE: Life Science Alliance Manuscript #LSA-2020-00944-T

Fatima Smagulova
Inserm/IrsetU1085

Dear Dr. Smagulova,

Thank you for submitting your manuscript entitled "In utero exposure to estrogenic pesticide chlordecone affects key epigenetic histone modifications and activates LINE-1 in human umbilical cord blood.". We would be happy to publish your paper in Life Science Alliance pending final revisions necessary to meet our formatting guidelines.

We apologize for the extended and unusual delay in getting back to you, which was caused due to a difficulty in chasing one of the reviewers assigned to the study. We are happy to report that the reviewers found the findings quite intriguing and have mainly only asked for clarifications that should be easily addressable with text revisions. We would, thus, like to accept this study with minor revisions at this stage and encourage you to submit a final version for publishing back to LSA.

Along with the points listed below, please also attend to the following:

- please consult our manuscript preparation guidelines <https://www.life-science-alliance.org/manuscript-prep> and make sure your manuscript sections are in the correct order
- please upload your main and supplementary figures as single files
- please add a Category and a Summary Blurb/Alternate Abstract for your manuscript in our system
- please add Author Contributions for your manuscript in our system
- please add your main, supplementary figures, and table legends to the main manuscript text after the references section
- we encourage you to revise the figure legends for figure 1 such that the figure panels match the actual figure (i.e. there is mentioning of panel A although there is no A panel in the actual figure)
- rename "Declaration of interest" section to "Conflict of interest"
- please upload your main manuscript text as an editable doc file;
- please upload your Supplementary Tables in editable .doc or excel format
- please check callouts for Figure 6C, D in your manuscript text (there are callouts for Figure 7C and D)
- please add callouts for Figures S4A, B, and S5A, B to your main manuscript text
- please add scale bars to Figure 3A
- please re-label the 'Sequencing data' section on page 25 as the Data Availability section and provide us the accession numbers for these data in the manuscript

A. FINAL FILES:

B. MANUSCRIPT ORGANIZATION AND FORMATTING:

Sincerely,

Shachi Bhatt, Ph.D.

Executive Editor

Life Science Alliance

<https://www.lsjournal.org/>

Interested in an editorial career? EMBO Solutions is hiring a Scientific Editor to join the international Life Science Alliance team. Find out more here -

https://www.embo.org/documents/jobs/Vacancy_Notice_Scientific_editor_LSA.pdf

Reviewer #1 (Comments to the Authors (Required)):

This is an interesting manuscript that focuses on CD exposure in a cohort but also cell culture based exposures to determine the effects on histone marks.

There are several aspects that need to be addressed:

1) The abstract needs to be revised to contain information about details of the study..eg the fact that the authors looked both at clinical samples as well as cell culture is not clear until very late in the manuscript. This needs to be included in the abstract (including providing the number of subjects from the Timoun cohort) and a statement that cells are used in the study

2) throughout the manuscript, there are several places where single sentences should be merged to generate paragraphs

3) the authors need to explain the value of their analysis in cord blood..eg what is the meaning of the study in cord blood versus target tissue

4) the title should be clearer that this is both a clinical assessment and in vitro and make it clear that the n of the clinical sample is quite small (e.g. better to state that this is a pilot study)

Reviewer #2 (Comments to the Authors (Required)):

Overall, this is an interesting manuscript that provides substantial information about histone modification and DNA methylation in a human population exposed to chlordecone.

1. These samples come from mixed cell populations. No mention is made about accounting for this. How do you know that your results are not confounded by changes in cell populations?

2. In the results sections, currently, it seems that ChiP-seq was performed from these structurally preserved nuclei. However, it seems that a different procedure was used as there was no nuclear extraction described in the methods section. Please update the way these samples are described

in the results to reflect the samples that were used.

3. CHIP from frozen tissue is notoriously difficult because formation of crystals disrupts the interactions between histones and DNA, as well as leading to DNA breakage. Were additional cryopreservation methods used to stabilize these interactions and the DNA? If not were methods used to isolate viable cells from those that were already dead? If no, do the authors think that there is potential confounding in the CHIP results from the type of sample used? Could this explain some of the differences observed between the results in cell lines and those in humans?

Dear Shachi Bhatt,

Thank you very much for the positive feedback! We are thankful for anonymous Reviewers for their comments and suggestions. We carefully revised our manuscript and have addressed all the concerns raised by the Referees. We updated the Title, Materials and Methods, Discussion and Figure legends sections. The changes in manuscript are marked in blue. The sequencing data were submitted to GEO databank, the accession number is under progress (usually it takes 1 day). We will add the accession number at the moment of the proof. Will be OK?

Thank you very much for considering our manuscript in your journal!

Thanking you,
Sincerely,
Fatima Smagulova

Reviewer #1 (Comments to the Authors (Required)):

This is an interesting manuscript that focuses on CD exposure in a cohort but also cell culture based exposures to determine the effects on histone marks.

Thank you very much for the kind words!

There are several aspects that need to be addressed:

1) The abstract needs to be revised to contain information about details of the study..eg the fact that the authors looked both at clinical samples as well as cell culture is not clear until very late in the manuscript. This needs to be included in the abstract (including providing the number of subjects from the Timoun cohort) and a statement that cells are used in the study.

We added to the Abstract of the manuscript, the information that cell culture studies were carried out.

2) throughout the manuscript, there are several places where single sentences should be merged to generate paragraphs

We merged the orphan sentences to the neighboring paragraph.

3) the authors need to explain the value of their analysis in cord blood..eg what is the meaning of the study in cord blood versus target tissue

Thank you for this comment! Indeed, we analysed the umbilical cord blood for studying the effects of CD exposure, however, the direct targets of CD are germline tissues such as testis and ovary, and somatic tissues such as liver, prostate and brain. Considering that exposure occurred during early embryonic period when tissue layers start to form, the common progenitors of many tissues gets affected, so the effects could be detected in many tissues, including blood. For example, a strong correlation between gestational Bisphenol A exposure and BDNF DNA methylation was observed in the umbilical cord blood of children from Columbia Center for Children's Environmental Health cohort (Kundakovic et al. 2015, 2015). The alteration in DNA methylation was correlated with brain and blood as was proved by analysis of postmortem brain (Davies et al. 2012).

We added the following statement in the discussion part, "is it relevant to use umbilical cord blood for analysis of prenatal effects of chlordecone. In a previous study, a strong correlation between gestational Bisphenol A exposure and BDNF DNA methylation in the umbilical cord blood was observed in the children from Columbia Center for Children's Environmental Health cohort (Kundakovic et al. 2015). The alteration in DNA methylation at promoters correlated in blood and brain as was proved by analysis of postmortem brain (Davies et al. 2012). The CD exposure occurred during entire early embryonic period, including early embryonic stage, when the tissue layers start to form, so common progenitors of many tissues could be affected, so the effects could be observed in many tissues, including blood".

4) the title should be clearer that this is both a clinical assessment and in vitro and make it clear that the n of the clinical sample is quite small (e.g. better to state that this is a pilot study)

Unfortunately, the format of title is very short (100 characters including spaces), we are not able to add the information of cell line used in our study.

Reviewer #2 (Comments to the Authors (Required)):

Overall, this is an interesting manuscript that provides substantial information about histone modification and DNA methylation in a human population exposed to chlordecone.

Thank you for the kind words, it is encouraging!

1. These samples come from mixed cell populations. No mention is made about accounting for this. How do you know that your results are not confounded by changes in cell populations?

Thank you for your comment! Indeed, we used whole umbilical cord blood and changes could arise from cell population changes. To address the question of cell population homogeneity in treated samples and control, we performed the analysis of cellular markers as was suggested in a recent study (Bakulski et al. 2016). In blood, there are several cell types with the granulocyte and lymphocytes being the major cells in umbilical cord blood. Since, each cell type have cell-specific marker expression it is feasible to verify the proportion of given cell type. For example, there are a large number of commonly used macrophage markers such as CD14, CD16, CD64, CD68, CD71 and CCR5. We detected in our ChIP-seq data that cell-specific markers for different cell types have no altered promoter H3K4me3 occupancy in control and exposed samples. We added this information in Supplementary Figure x and a statement has also been included in the Materials and Methods part. For immunofluorescence analysis, we chose cells with similar nucleus morphology and considering that highest cell population are granulocyte and lymphocytes, our results reflect changes in these major cell populations. We also added this statement in the Materials and Methods part.

2. In the results sections, currently, it seems that ChIP-seq was performed from these structurally preserved nuclei. However, it seems that a different procedure was used as there was no nuclear extraction described in the methods section. Please update the way these samples are described in the results to reflect the samples that were used.

Thank you for this comment. Indeed, we used structurally preserved nuclei for ChIP as well. For ChIP, the centrifugation at 900g during 10 min precipitate mainly intact nuclei, we verified the cell suspension under microscope and we observed the majority of nuclei are intact. Partially broken nuclei were eliminated in the next step by washing with low detergent solution and sedimentation at 900g, the broken nuclei will not precipitate at this speed.

3. ChIP from frozen tissue is notoriously difficult because formation of crystals disrupts the interactions between histones and DNA, as well as leading to DNA breakage. Were additional cryopreservation methods used to stabilize these interactions and the DNA?

Normally the frozen tissue also preserves well the nucleosome, as we observed in our previous experiments with animal tissue. Our ChIP-seq data are reproducible between replicates, and in general, we have low variation between samples. Histones are small charged proteins and in general, they are well preserved during freeze and thaw. To avoid any protein degradation, we also immediately crosslinked cells with paraformaldehyde.

If not were methods used to isolate viable cells from those that were already dead? If no, do the authors think that there is potential confounding in the ChIP results from the type of sample used?

Yes, the ideal way is to use a fresh blood samples. But it is not feasible to obtain fresh blood in human studies from cohorts as biological sample collection requires the time. We assume that reproducibility in biological replicates could reflect the tissue homogeneity and preservation, as and in degraded tissue, results are not that reproducible.

Could this explain some of the differences observed between the results in cell lines and those in humans?

We rather believe, some small changes between cell line and blood could be explained by the fact that cell composition in umbilical cord is not the same as in cell line. Besides that, in vivo samples have difference due to influence from other organs, e.g. hormones and secreted factors from tissues such hormonal tissues.

References:

- Bakulski KM, Feinberg JI, Andrews SV, Yang J, Brown S, L McKenney S, Witter F, Walston J, Feinberg AP, Fallin MD. 2016. DNA methylation of cord blood cell types: Applications for mixed cell birth studies. *Epigenetics* 11: 354–362.
- Davies MN, Volta M, Pidsley R, Lunnon K, Dixit A, Lovestone S, Coarfa C, Harris RA, Milosavljevic A, Troakes C, et al. 2012. Functional annotation of the human brain methylome identifies tissue-specific epigenetic variation across brain and blood. *Genome Biol* 13: R43.
- Kundakovic M, Gudsnuk K, Herbstman JB, Tang D, Perera FP, Champagne FA. 2015. DNA methylation of BDNF as a biomarker of early-life adversity. *Proc Natl Acad Sci U S A* 112: 6807–6813.

March 29, 2021

RE: Life Science Alliance Manuscript #LSA-2020-00944-TR

Dr. Fatima Smagulova
Inserm/IrsetU1085
9 avenue du Prof. Léon Bernard
Rennes 35000

Dear Dr. Smagulova,

Thank you for submitting your Research Article entitled "In utero exposure to chlordecone affects histone modifications and activates LINE-1 in cord blood.". It is a pleasure to let you know that your manuscript is now accepted for publication in Life Science Alliance. Congratulations on this interesting work.

Your manuscript will now progress through copyediting and proofing. It would be okay to add the GEO accession numbers at the proofs stage. It is journal policy that authors provide original data upon request.

DISTRIBUTION OF MATERIALS:

Again, congratulations on a very nice paper. I hope you found the review process to be constructive and are pleased with how the manuscript was handled editorially. We look forward to future exciting submissions from your lab.

Sincerely,

Shachi Bhatt, Ph.D.

Executive Editor

Life Science Alliance

<https://www.lsjournal.org/>
